# Hyperandrogenic Symptoms Are a Persistent Suffering in Midlife Women with PCOS; a Prospective Cohort Study in Sweden

**DOI:** 10.3390/biomedicines11010096

**Published:** 2022-12-30

**Authors:** Sofia Persson, Kumari A. Ubhayasekera, Jonas Bergquist, Sahruh Turkmen, Inger Sundström Poromaa, Evangelia Elenis

**Affiliations:** 1Department of Women’s and Children’s Health, Uppsala University Hospital, 751 85 Uppsala, Sweden; 2Department of Chemistry—BMC, Analytical Chemistry and Neurochemistry, Uppsala University, 751 24 Uppsala, Sweden; 3Department of Clinical Sciences, Obstetrics and Gynecology, Sundsvalls Research Unit, Umeå University, 901 85 Umeå, Sweden; 4Reproduction Centre, Women’s Clinic, Uppsala University Hospital, 751 85 Uppsala, Sweden

**Keywords:** PCOS, PCOS phenotypes, hyperandrogenism

## Abstract

Polycystic ovary syndrome (PCOS) is a common endocrine disorder among women, and the majority suffers from hyperandrogenism. Hyperandrogenism causes psychological morbidity and impaired quality of life in women with PCOS during the reproductive years, but data on prevalence and impact during midlife are lacking. Thus, this study aimed to address whether hyperandrogenism persists into midlife and, if so, what impact it has on quality of life. In order to answer this question, we performed a multicenter prospective cohort study, where we included women already diagnosed with PCOS who had reached the age of 45 years or more and age-matched controls. All participants underwent a physical exam, structured medical interview, biochemical testing and filled out self-assessment questionnaires. More than 40% of the women with PCOS and 82% of those who presented with the hyperandrogenic phenotype at the diagnostic work-up still suffered from hirsutism. Circulating testosterone levels were similar between women with PCOS and controls while free androgen index was higher in women with PCOS, independent of weight. Women with hyperandrogenic PCOS expressed persisting concerns regarding hirsutism at the follow-up assessment. In conclusion, women with PCOS who present with hyperandrogenic symptoms at the time they are diagnosed with PCOS have a higher risk of persistent androgenic symptoms and impaired quality of life in midlife.

## 1. Introduction

Polycystic ovary syndrome (PCOS) is a common endocrine condition that affects 8 to 13% of women of reproductive age [1]. The syndrome consists of three features in various combinations: hyperandrogenism (HA), anovulation and ultrasonographic findings of polycystic ovaries [2]. Based on the combinations of these criteria, four different PCOS phenotypes are identified [3]. Three of the phenotypes include hyperandrogenism, and these phenotypes are also considered the most severe [4,5]. The proportion of women with the different phenotypes varies with age, with a decrease in prevalence of hyperandrogenic phenotypes over time and an increase in the prevalence of the ovulatory phenotype, as well as the proportion of women who no longer fulfil the criteria for PCOS [6,7]. At midlife, women with PCOS seem to continue to have higher serum androgen levels than women without the condition and their degree of clinical hyperandrogenism seems to extend at least until the age of 50 [7,8]. On top of that, menopause may be considered as a state of relative androgen excess in general [9], when even non-PCOS women might experience mild androgenic symptoms such as hirsutism [10]. This is mainly due to a decline in estrogen- and sex hormone-binding globulin (SHBG) production [9,11,12,13], consequently leading to a relative increase in free androgen levels [14].

PCOS symptoms are a major cause of psychological morbidity and have a negative impact on women’s health-related quality of life (HRQoL) [15]. However, the existing studies on quality of life among PCOS women only include limited samples and mainly focus on women of reproductive age. In fact, in a systematic review by Behboodi et al. from 2018, only 4 out of 52 studies included women with PCOS aged over 40 years [15]. Although clinical experience suggests that midlife women with PCOS still suffer from hyperandrogenic symptoms that affect their quality of life, little is known on the topic [16].

The aim of our study was to compare the presence of hyperandrogenic symptoms, testosterone and SHBG levels between women with and without PCOS above 45 years of age, and to study the impact of hyperandrogenism on the overall health-related quality of life in women with normoandrogenic and hyperandrogenic POCS phenotypes.

The research questions were: Are hyperandrogenic symptoms more prevalent among midlife women with PCOS than among age-matched controls, and are women with hyperandrogenic PCOS phenotypes at the time of diagnosis more affected than women who were normoandrogenic at the time of diagnosis? Are there any differences in testosterone levels and free androgen index between midlife women with and without PCOS? Is quality of life worse in midlife in women with PCOS who presented with the hyperandrogenic phenotype at the time of diagnosis in comparison with those who were normoandrogenic?

## 2. Materials and Methods

The study was designed as a multicenter prospective cohort study set at the departments of Women’s and Children’s Health at Uppsala University Hospital (Uppsala), Sundsvall County Hospital (Sundsvall), Karolinska University Hospital (Stockholm) and Falun County Hospital (Falun). Study inclusion period lasted between January 2017 and December 2021, with an almost 18-month temporary stop during the COVID pandemic.

### 2.1. Study Population

We identified women with PCOS through the outpatient register at the collaborating clinics and sent out letters of invitation for participation in the study. Inclusion criteria for the study group were prior diagnosis of PCOS according to the Rotterdam or NIH criteria and present age of above 45 years. All women with a past PCOS diagnosis were considered, irrespective of whether they still fulfilled PCOS criteria or not at the time of assessment. Women with PCOS were classified as having a normoandrogenic or hyperandrogenic phenotype based on hyperandrogenic symptoms and androgen levels when first diagnosed with the condition, identified either in the woman’s medical record or according to patient recall. If medical records were available, we followed the Rotterdam criteria for classification. In the few cases where medical records were not available, or lacked relevant information on PCOS features/androgen levels, we relied on patient recall; if participant stated that unwanted body hair or acne were the main causes for seeking health care at the time of PCOS diagnosis, they were classified as having the hyperandrogenic (HA) phenotype of PCOS. Women with irregular menstrual periods or anovulatory infertility but no hyperandrogenic symptoms were classified as having the normoandrogenic (NA) phenotype.

Controls were matched for age and area of residence and identified in the population register of Uppsala County and Västernorrland County, respectively. Among the identified eligible women, one control per PCOS woman was randomly selected and invited. Due to the difficulties of recruiting controls during the pandemic, adjustments in the study protocol were made by extending the inclusion to a well-defined cohort of controls recruited from a previous study performed in 2006–2007 [17] using the same inclusion criteria as above. None of the controls had a prior PCOS diagnosis in their medical records. In addition, at inclusion, we confirmed that controls had no prior history of oligomenorrhea/amenorrhea, hirsutism or anovulatory infertility.

Exclusion criteria for all participants were inability to understand Swedish or attend the health care exam in-person.

### 2.2. Structured Medical History and Physical Exams

All women who consented to the study participated in a follow-up assessment, which included structured medical history, a health examination and blood sampling at the gynaecological department of the participating clinics. The structured medical interview included previous and ongoing health problems, ongoing medication, menstrual history, ongoing hormonal treatment (such as hormonal contraception or menopausal hormonal treatment (MHT)) and previous oophor- or hysterectomy. Women were asked about hair removal in the last 12 months (yes/no), frequency of hair removal (daily/less than daily), current bothersome acne and ongoing acne treatments. The menopausal status was variable among participants (pre-menopausal or postmenopausal, with or without hormonal treatment). Women with at least one menstrual period and no hormonal treatment during the past year were classified as pre-menopausal. Women with amenorrhea in the preceding 12 months and lack of on any type of hormonal treatment were classified as post-menopausal. Women who were on hormonal treatment (combined oral contraceptives (COC), hormonal intrauterine device (IUD), oral progestins or MHT) were classified as (i) post-menopausal if they had an AMH < 0.1 pmol/l or (ii) unclassified if AMH was not measured and (iii) premenopausal if AMH was >0.1 pmol/L. Participants were weighed on an electronic scale in light clothes and without shoes, to the nearest 0.1 kg. Height without shoes was measured to the nearest centimeter.

### 2.3. Biochemical Analysis by Supercritical Fluid Chromatography–Tandem Mass Spectrometry

Blood samples were available in 118 women with PCOS and 39 controls. Blood samples were centrifuged and stored in −70 °C until analyses. Testosterone was analyzed with Supercritical Fluid Chromatography (Waters ACQUITY^®^ UPC2™, Milford, MA, USA) coupled with tandem mass spectrometry (XEVO^®^ TQ-S, Milford, MA, USA). In brief, the analysis was performed with a BEH column (150 mm 3.0 mm, 1.7 µm) at 40 °C (Waters, Milford, MA, USA). All data collected in centroid mode were obtained using MassLynx NT4.1 software (Waters, Milford, MA, USA). The method has previously been validated, and the limit of quantification (LOQ) is 0.1 nmol/L [18]. In brief, 100 µL of serum samples were spiked with 50 µL of ^13^C3- testosterone and protein precipitation was carried out with 0.1% BHT in ACN. Samples were vortexed and centrifuged for 15 min at 13,000 g at 4 °C. Each sample’s supernatant was collected, and a TurboVap^®^ Evaporator was used to evaporate the solvent under nitrogen at 35 °C (Biostage, Uppsala, Sweden). The samples were resuspended in 100 µL of methanol and 1µL was injected during the analysis [19].

Serum concentrations of SHBG were analyzed by solid-phase chemiluminescent immunometric assays. Serum AMH levels were measured using the fully automated Access Dxi sandwich immunoassay (B13127, Beckman Coulter, Brea, CA, USA). This assay measures the proAMH and the cleaved AMH_N,C_ complex and uses recombinant human AMH as a calibrator. SHBG and AMH were analyzed by the accredited Clinical Chemistry laboratory at Uppsala university hospital. Free testosterone was calculated as total testosterone/SHBG ratio and hyperandrogenism was defined as a ratio above 0.05.

### 2.4. Outcomes

Three hyperandrogenic symptoms were evaluated at the follow-up assessment; hirsutism, androgenic alopecia and acne. We used the modified Ferriman–Gallwey (mFG) score to assess severity of hirsutism. Since an accepted cut-off value of the mFG score for Scandinavian women is lacking, we considered a total score of 8 or above as indicative of androgen excess [20,21,22]. Androgenic hair loss was estimated based on the clinician’s evaluation of the Ludwig scale, where Stage I means hair thinning on top of the head, and Stage II means the scalp is starting to show. In Ludwig Stage III, the scalp is very noticeable, and the hair left is so thin it does not conceal the scalp [21]. Information on current bothersome acne and treatment of acne were self-reported by the study participants. In women with PCOS, the quality of life was measured by the PCOS Health-Related Quality of Life questionnaire (PCOSQ) [23]. The questionnaire has been translated to and validated in Swedish [24]. The PCOSQ contains five domains: (i) Emotional concerns such as depressive symptoms, worries and low self-esteem as a result of having PCOS (8 items), (ii) Hirsutism (5 items), (iii) Weight (5 items), (iv) Infertility (4 items) and (v) Menstrual Disorders (4 items). Each item can be answered by choosing one of seven different options, with responses ranging from 1 (poorest function) to 7 (optimal function) [15,23]. A median score for each domain was calculated and used in the analyses.

### 2.5. Statistics

The following covariates were recorded and used in the analysis: age (years), body mass index (BMI) (categorised according to the WHO classification [25], i.e., normal weight BMI < 25.0 kg/m^2^, overweight 25.0–29.9 kg/m^2^, and obesity BMI ≥ 30.0 kg/m^2^), marital status (single/cohabiting/married). Categorical variables were compared by Chi-square test or the Fischer’s exact probability test (when the lowest frequency in any of the cells was below 5). Continuous variables were compared by Mann–Whitney U-test, as some of the variables did not follow the normal distribution. For each variable, we drew two sets of comparisons: (i) between women with and without PCOS, and (ii) between women with NA PCOS and HA PCOS phenotypes. Multiple linear regression was used to assess the association between BMI and PCOS phenotypes on hirsutism, testosterone and FAI. The IBM Statistical Package for Social Sciences (SPSS) version 26 (IBM Inc., Armonk, NY, USA) was used for the statistical analyses. A two-sided *p*-value < 0.05 was regarded as statistically significant. Missing values were not imputed.

## 3. Results

### 3.1. Characteristics of the Study Population and Anthropometry

In total, 124 women with PCOS and 74 controls were included in the study. Demographic variables are shown in Table 1. The median age was 50 years in women with PCOS and 51 years in controls. The majority of participants were born in Sweden or the Nordic countries. Women with PCOS had a higher median BMI and a larger proportion of them were obese compared to controls (47.2% vs. 16.2%, *p* < 0.001). Most women did not use any hormonal contraception and among those who did, the hormonal IUD was the most common. Fewer women with PCOS (27.4%) than controls (39.2%, *p* = 0.013) were postmenopausal, and 3.2% of women with PCOS and 6.8% of controls were currently on MHT (*p* = 0.296).

### 3.2. Hyperandrogenic Symptoms and Testosterone Levels

In Table 2, hyperandrogenic symptoms and testosterone levels are displayed. According to the mFG score, more than 40% of women with PCOS still suffered from hirsutism, whereas the corresponding proportion among controls was around 4% (*p* < 0.001). As expected, almost all midlife women with hirsutism originated from the HA PCOS subgroup, with 82% of women with HA PCOS reporting persisting symptoms at midlife. The vast majority of women with PCOS reported removing facial hair during the past 12 months, with 45.2% of HA PCOS and 19.4% of NA PCOS doing so on a daily basis (*p* < 0.001). In contrast, the majority of controls did not remove any facial hair during the past year (79.7%). Androgenic hair loss was more common in women with PCOS, and more severe in women with HA PCOS than in women with the NA phenotype, while it was uncommon in the controls. A third of women with PCOS had a current problem with acne in contrast to one sixth of controls (*p* = 0.006).

The circulating total testosterone levels were similar in women with PCOS and controls (*p* = 0.102) with a tendency towards higher testosterone in the PCOS HA phenotype compared to the NA phenotype (*p* = 0.079). Women with PCOS had lower SHBG and higher FAI than controls (*p* < 0.01). Within the PCOS group, FAI was higher in the HA phenotype than in the NA phenotype (*p* = 0.028), Table 3. In further analyses stratified by BMI, testosterone was higher in normal-weight women with PCOS compared with controls (0.34 vs. 0.27 ng/mL, *p* = 0.012). However, testosterone levels did not differ among overweight or obese women with PCOS and controls (*p* = 0.791), Appendix A. SHBG was similar between normal-weight women with or without PCOS (*p* = 0.148) but was lower in overweight or obese women with PCOS than in controls (*p* < 0.004), Appendix A.

In the multiple linear regression analyses, mFG score was positively associated with the HA PCOS phenotype (*p* < 0.001) and obesity (*p* = 0.027). Testosterone was positively associated with overweight (*p* < 0.026) but not with obesity (*p* = 0.157) or PCOS phenotype. FAI was positively associated with the HA PCOS phenotype (*p* = 0.042), overweight (*p* = 0.008) and obesity (*p* < 0.001).

### 3.3. Quality of Life among Women with PCOS

The PCOSQ health quality scores are presented in Table 4. Women with PCOS had persisting concerns regarding hirsutism (median score 4.7) and weight (median score 3.2). Women with HA PCOS had greater concerns regarding hirsutism than women with NA PCOS (median score 3.0 in HA PCOS vs. 5.8 in NA PCOS women, *p* < 0.001). Infertility was no longer considered a problem, while menstrual disturbances and emotional concerns due to PCOS were still a source of mild suffering, Table 4.

## 4. Discussion

In our study examining peri- and postmenopausal women with PCOS and controls, we found that hirsutism persisted and constituted a significant source of psychological suffering for midlife women with PCOS but not for controls. As part of the normal menopausal process, testosterone production from the ovaries and adrenals decreases while SHBG secretion from the liver falls [26]. These changes lead to stable or even increased levels of free circulating androgens after menopause in all women [9]. Previous research has shown that this relative androgen excess is more pronounced among postmenopausal women with PCOS compared to women without the condition, which is in line with our findings [7,27,28]. According to our results, testosterone levels were similar between controls and women with PCOS, with a tendency towards higher values in the hyperandrogenic PCOS phenotype. However, since our sample size was small, the results might be different in a larger cohort.

It has been reported before that women with PCOS reaching menopause have a time-dependent trend with decrease in androgens at first and a testosterone increase after the age of 50 [8]. Despite the persisting biochemical hyperandrogenism, it is generally expected that the signs and symptoms of PCOS would become ameliorated with ageing [29]. In contrast, our findings suggest that a substantial proportion of women with PCOS still suffer from hirsutism, despite having reached or surpassed menopause, and there is an unmet need for treatment since current guidelines lack targeted treatment options in midlife women [21]. Among women with PCOS who presented with hyperandrogenic symptoms at the time of their diagnostic work-up, 80% were still hirsute at follow up and 45% needed daily hair removal. One contributing factor to these findings could be that four out of five (81.5%) of women with PCOS were overweight or obese and almost half of women with PCOS had a BMI above 30 kg/m^2^. Obesity causes functional hyperandrogenism among women, irrespectively of PCOS status, mainly by lowering SHBG-production from the liver or increasing insulin resistance, leading to elevated circulating free androgens [30,31], also noted in our sample. However, when focusing on normal-weight women only, we observed that testosterone and FAI were higher in women with PCOS compared to controls. All of the latter imply that overweight/obesity does not alone explain hyperandrogenism in midlife women with PCOS.

Women with the HA phenotype reported body hair as still being a problem of moderate significance, whereas women with the NA PCOS phenotype were not equally concerned by it. Despite their suffering, relatively few midlife women with HA PCOS reported being on MHT or COC at the time of assessment. While COCs are first-line treatment for hyperandrogenism in women of reproductive age [21], this option is not appropriate for PCOS women in midlife due to the inherent risk for thromboembolism. Moreover, although systemic MHT could be an alternative, it remains to be seen whether the associated risks outweigh the expected benefits [32]. Other anti-androgenic agents such as spironolactone and cyproterone acetate, often used in conjunction with COCs, are off label in PCOS and have not been tested in women of advanced age [21]. Unfortunately, we lack information on anti-androgenic treatment other than COC or MHT in the present study.

In addition, a substantial proportion of PCOS women reported that their weight had a negative impact on their quality of life. Studies in infertile women with PCOS report that infertility combined with hirsutism and weight gain have strong negative impact on quality of life [33,34], and our results imply that the impact of hirsutism and overweight or obesity on quality of life persist beyond the fertile age. Obesity is a risk factor for several medical conditions and contributes to a worsening of hyperandrogenism, which could be an explanation to the negative impact on quality of life [10]. On the positive side, the majority of our midlife women with PCOS were generally no longer concerned about infertility or menstrual disturbances.

Through this study, we have identified both a problem (i.e., persistence of androgenic symptoms in PCOS women around menopausal age, especially among those with the hyperandrogenic phenotype) and an unmet need for treatment of symptoms. Efforts should therefore be made to alleviate the suffering of a great proportion of women.

### 4.1. Comparison to Other Studies

There is a scarcity of studies on women with PCOS around menopause in regard to the prevalence of androgenic symptoms. Nevertheless, our findings are consistent with the few that are at hand. In the longest prospective study of PCOS, a cohort of women with PCOS and age- and BMI-matched controls were followed from 1987 until 2019 [35]. At twenty-year follow-up, total testosterone and FAI had declined in both groups, but were still higher in women with PCOS than in controls, with two thirds of them still considered hirsute [36]. At thirty-year follow-up, women with PCOS were still hirsute to a higher degree than controls, despite having similar BMI. Furthermore, testosterone, SHBG and FAI no longer differed between the groups [36]. A cross-sectional study of women above 55 years of age yielded similar results; the study included 20 postmenopausal women with PCOS and equal number of BMI- and age-matched controls [27]. Women with PCOS exhibited a higher level of androgens of both adrenal and ovarian origin compared to controls [27]. Unfortunately, none of the studies analyzed the PCOS phenotypes separately. In contrast, a cohort study demonstrated a shift in the distribution of PCOS phenotypes over time, with more women gaining menstrual cyclicity and fewer women exhibiting hyperandrogenism within the older group of women with PCOS [7].

### 4.2. Strengths and Limitations

The principal study strength is its prospective research design; study subjects were objectively assessed and examined by the research personnel, limiting the risk of information bias. We also had access to medical records and were able to ask follow-up questions and clarifications, decreasing the misclassification risk of the controls. Furthermore, study recruitment focused on women with PCOS in the perimenopausal period, an important stage in life where studies on women with PCOS are lacking. Moreover, the assessment of hirsutism and alopecia was based on the Ferriman–Gallwey and Ludwig scores, which were assessed by healthcare professionals. In addition, we collected information on hyperandrogenic symptoms at the time of PCOS diagnosis, either by access to the participants’ medical records or by the women’s recall. Lastly, we were the first to explore the hyperandrogenic symptoms among PCOS women after taking into account their PCOS phenotype.

Study limitations constitute the small sample size and the difficulties to reach the aimed number of participants especially of control women. This could be partly explained by the coronavirus (COVID-19) pandemic and the related precautions, resulting in a longer inclusion period and a lower participation rate. Furthermore, we may have miss-classified some women with the NA PCOS phenotype. Among the cases where we relied on self-reporting for phenotype classification, it is possible that non-hirsute women were unaware of biochemical hyperandrogenism at the time of their diagnosis. In addition, we cannot exclude the risk of recall bias, which is however deemed to be low since we had access to the medical records for the majority of PCOS women and controls. Lastly, analysis of dehydroepiandrosterone sulfate (DHEAS) and other androgens at the follow-up assessment could have contributed to an overall estimation of biochemical hyperandrogenism in midlife PCOS [2]. Our lack of DHEAS analysis may limit the comparison to other studies, and it also hinders analysis of relation between DHEAS and health outcomes.

## 5. Conclusions

Women with PCOS who present with hyperandrogenic symptoms at the time of PCOS diagnosis have a higher risk of persistent androgenic symptoms and impaired quality of life in midlife than those without hyperandrogenic symptoms. At the same time, very few had treatment for their symptoms. Women with PCOS should be made aware of their health risks and treatment options and physicians should do their outmost to alleviate their symptoms not only during the reproductive period, but also later in life.

## Figures and Tables

**Table 1 biomedicines-11-00096-t001:** Demographic information of study population.

	Controls	PCOS
	*n* = 74	*n* = 124
	*n* (%)	*n* (%)
Age, median (min, max)	51 (45, 66)	50 (45, 66)
Marital status		
Married	50 (73.5)	79 (79.0)
Single household	16 (23.5)	20 (20.0)
Other	2 (2.9)	1 (1.0)
BMI, median (min, max)	24.9 (18.2, 41.4)	29.7 (21.2, 50.0)
<25.0 kg/m^2^	37 (50.0)	22 (17.9)
25.0–29.9 kg/m^2^	25 (33.8)	43 (35.0)
≥30.0 kg/m^2^	12 (16.2)	58 (47.2)
Contraceptives		
No hormonal method	58 (79.5)	94 (75.8)
Combined oral contraception	1 (1.4)	2 (1.6)
Hormonal IUD	14 (19.2)	18 (14.5)
Oral progestins	0 (0)	10 (8.1)
MHT		
No	68 (93.2)	120 (96.8)
Yes	5 (6.8)	4 (3.2)
Menopausal status ^1^		
Premenopausal	34 (45.9)	82 (66.1)
Postmenopausal	29 (39.2)	34 (27.4)
Unclassified	11 (14.9)	8 (6.5)
Country of birth		
Nordic countries	71 (95.9)	105 (84.7)
Other countries	3 (4.1)	5 (4.0)
Not reported	0 (0.0)	14 (11.3)

^1^ Defined in the following way: premenopausal = bleeding within the last 12 months and no hormonal treatment or hormonal treatment and AMH ≥ 0.1 pmol/L; postmenopausal = no bleedings within the last 12 months, no hormonal treatment or hormonal treatment and AMH < 0.1 pmol/L; unclassified = no bleedings within the last 12 months, hormonal treatment and AMH absent or bleedings within the last 12 months with hormonal treatment and AMH absent. IUD = intrauterine device; MHT = menopausal hormone therapy. Percentages are presented as % of subjects with a reported value.

**Table 2 biomedicines-11-00096-t002:** Androgenic symptoms, treatment and biochemical hyperandrogenism at the follow-up assessment in women with PCOS and controls.

	Controls *n* = 74	PCOS *n* = 124	*p*-Value ^1^	NA PCOS *n* = 62	HA PCOS *n* = 62	*p*-Value ^2^
	*n* (%)	*n* (%)		*n* (%)	*n* (%)	
Hair removal last 12 months			<0.001			<0.001
No	59 (79.7)	43 (34.7)		32 (51.6)	11 (17.7)	
Less than daily	14 (18.9)	41(33.1)		18 (29.0)	23 (37.1)	
Daily	1 (1.4)	40 (32.3)		12 (19.4)	28 (45.2)	
Androgenic hair loss			0.066			0.190
No	41 (95.3)	96 (77.4)		52 (83.9)	44 (71.0)	
Ludwig grade I	2 (4.7)	21 (16.9)		9 (14.5)	12 (19.4)	
Ludwig grade II	0 (0)	6 (4.8)		1 (1.6)	5 (8.1)	
Ludwig grade III	0 (0)	1 (0.8)		0 (0)	1 (1.6)	
Acne, current problem			0.006			0.697
No	64 (86.5)	86 (69.4)		42 (67.7)	44 (71.0)	
Yes	10 (13.5)	38 (30.6)		20 (32.3)	18 (29.0)	
Acne treatment, ongoing			0.375			0.619
No	71 (95.9)	120 (96.8)		59 (95.2)	61 (98.4)	
Yes	3 (4.1)	4 (3.2)		3 (4.8)	1 (1.6)	
Ferriman–Gallwey score			<0.001			<0.001
0–7	71 (95.9)	71 (57.7)		60 (96.8)	11 (18.0)	
≥8	3 (4.1)	52 (42.3)		2 (3.2)	50 (82.0)	
Biochemical hyperandrogenism ^3^						
Testosterone, ng/mL, median (min, max)	0.28 (0.13, 0.81)	0.34 (0.02, 1.29)	0.102	0.32 (0.02, 0.83)	0.36 (0.13, 1.29)	0.079
SHBG, nmol/L, median (min, max)	65.0 (27.0, 142.0)	41.0 (6.4, 110.0)	<0.001	43.0 (6.4, 108.0)	39.0 (9.6, 110.0)	0.321
FAI, median (min, max)	0.004 (0.00, 0.01)	0.008 (0.00, 0.06)	<0.001	0.006 (0.00, 0.05)	0.010 (0.00, 0.06)	0.028

Statistical analyses by chi^2^-test or Fischer’s exact test. Continuous variables compared by Mann–Whitney U-test. ^1^ Comparing controls and total PCOS, ^2^ Comparing PCOS phenotypes. ^3^ Blood samples were available in 118 women with PCOS and 39 controls. HA = hyperandrogenic; NA = normoandrogenic PCOS phenotypes.

**Table 3 biomedicines-11-00096-t003:** Variables influencing hirsutism, testosterone and free androgen index in women with PCOS and controls at the follow-up assessment based on multiple regression modelling.

	Difference in Mean	95% CI	*p*-Value
Ferriman–Gallwey score			
PCOS group			
Control	(Ref)		
NA PCOS	1.3	(−0.1, 2.6)	0.061
HA PCOS	9.1	(7.7, 10.5)	<0.001
BMI			
<25.0 kg/m^2^	(Ref)		
25.0–29.9 kg/m^2^	0.8	(−0.5, 2.1)	0.240
≥30.0 kg/m^2^	1.6	(0.2, 2.9)	0.027
Testosterone			
PCOS group			
Control	(Ref)		
NA PCOS	−0.001	(−0.079, 0.077)	0.984
HA PCOS	0.063	(−0.017, 0.143)	0.123
BMI			
<25.0 kg/m^2^	(Ref)		
25.0–29.9 kg/m^2^	0.081	(0.010, 0.152)	0.026
≥30.0 kg/m^2^	0.053	(−0.021, 0.126)	0.157
FAI			
PCOS group			
Control	(Ref)		
NA PCOS	0.002	(−0.001, 0.005)	0.194
HA PCOS	0.003	(0.000, 0.007)	0.042
BMI			
<25.0 kg/m^2^	(Ref)		
25.0–29.9 kg/m^2^	0.004	(0.001, 0.007)	0.008
≥30.0 kg/m^2^	0.007	(0.004, 0.009)	<0.001

The model is adjusted for age, menopause status, menopausal hormone treatment and combined oral contraceptive use. HA = hyperandrogenic; NA = normoandrogenic PCOS phenotypes; BMI = body mass index; FAI = free androgen index.

**Table 4 biomedicines-11-00096-t004:** Health-related quality of life in women with PCOS at the follow-up assessment.

PCOS-Q Domains	All PCOS Median (Min, Max)	NA PCOS Median (Min, Max)	HA PCOS Median (Min, Max)	*p*-Value ^1^
Emotional concerns	5.9 (2.8, 7.0)	6.1 (2.8, 7.0)	5.6 (2.8, 7.0)	0.095
Hirsutism	4.7 (1.0, 7.0)	5.8 (1.0, 7.0)	3.0 (1.0, 7.0)	<0.001
Weight concerns	3.2 (1.0, 7.0)	3.8 (1.0, 7.0)	3.0 (1.0, 7.0)	0.151
Infertility concerns	7.0 (1.5, 7.0)	7.0 (1.5, 7.0)	6.8 (2.5, 7.0)	0.175
Menstrual concerns	5.6 (2.0, 7.0)	5.8 (2.0, 7.0)	5.5 (3.0, 7.0)	0.634

^1^ Comparison between PCOS normoandrogenic (NA) and hyperandrogenic (HA) phenotypes, data analysed with Mann–Whitney U-test.

## Data Availability

The data presented in this study are available on request from the corresponding author. The data are not publicly available due to confidentiality.

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
