# Peer review of "Hyperandrogenic Symptoms Are a Persistent Suffering in Midlife Women with PCOS; a Prospective Cohort Study in Sweden"

_biomedicines, 2022, doi:10.3390/biomedicines11010096_

Round 1

Reviewer 1 Report

Thanks for the interesting study about PCO patients

I recommend publication

Additional pitfalls of control definition: “Controls were matched for age and area of residence and were randomly selected from the population”

Author Response

The authors thanks for the review and the recommendation for publication. We have now revised the manuscript and clarified how the controls were selected.

Reviewer 2 Report

The authors present the results of a multicenter prospective cohort study dedicated to age dynamics of hyperandrogenic symptoms in patients with PCOS and related quality of life.  The topic is of importance and the presented data increase the body of evidence regarding the persistence of hyperandrogenism in the midlife period associated with a decrease in quality of life.

The manuscript was reviewed using the STROBE Statement—Checklist of items that should be included in reports of cohort studies (https://www.equator-network.org/reporting-guidelines/strobe/).

The abstract provides an informative and balanced summary of what was done and what was found; the Introduction presents the scientific background and states objectives. The Methods section clearly describes the key elements of study design, settings, and study population. Importantly, the authors used only validated methods for hormone analysis as well as for quality-of-life assessment.

The comments and suggestions by sections:

Consider adding a flow diagram and race characteristics of study participants (if you included not only Caucasians, but also Asians etc.).

Indicate the number of participants with missing data for each variable of interest and describe how you treat missing data in the Statistical analysis section.

Please also consider mentioning the absence of data on DHEAS as a study limitation in the discussion section

There are some grammatical errors and typos in the manuscript.

The reviewer’s decision: Accept after minor revision (corrections to minor methodological errors and text editing)

Author Response

The authors thanks for the reviw and have revised the manuscript in line with the comments.

We considered to add a flow diagram over the study population, but we prefer to keep the information in Table 1, which was updated with country of birth. The majority of study participant originated from Sweden or other Nordic countries.

The information on missing data has now been described in the method section and is also discussed in the discussion section.

A section on the absence of DHEAS has been added to the study limitations.

The manuscript has been revised and edited for linguistic errors.